# The Role of Contrast-Enhanced Ultrasound (CEUS) in the Detection of Neoplastic Portal Vein Thrombosis in Patients with Hepatocellular Carcinoma

Lucia Cerrito [ID], Maria Elena Ainora, Silvino Di Francesco, Linda Galasso, Antonio Gasbarrini [ID] and Maria Assunta Zocco *[ID]

CEMAD Centro Malattie dell'Apparato Digerente, Fondazione Policlinico Universitario "A. Gemelli" IRCCS, Università Cattolica del Sacro Cuore Roma, Largo A. Gemelli, 8, 00168 Roma, Italy; lucia.cerrito@policlinicogemelli.it (L.C.); mariaelena.ainora@policlinicogemelli.it (M.E.A.); silvino.difrancesco@outlook.it (S.D.F.); linda.galasso0817@gmail.com (L.G.); antonio.gasbarrini@unicatt.it (A.G.)
* Correspondence: mariaassunta.zocco@unicatt.it; Tel.: +39-06-3015-6018; Fax: +39-06-3015-7232

**Abstract:** Hepatocellular carcinoma (HCC) is the principal primary liver cancer and one of the most frequent malignant tumors worldwide in patients with chronic liver disease. When diagnosed at an advanced stage, it is often associated with portal vein tumor thrombosis (PVTT), which heavily affects patients' prognosis. Imaging evaluation is crucial in PVTT detection and staging; computed tomography and magnetic resonance are the principal diagnostic tools. Contrast-enhanced ultrasound (CEUS) is a non-invasive and easily repeatable method that can also be used in patients with impaired renal function. It represents an important means for the identification of PVTT, particularly differentiating neoplastic and non-neoplastic thrombosis through the analysis of ultrasound enhancement characteristics of the thrombosis (arterial hyperenhancement and portal washout), thus allowing more refined disease staging, appropriate treatment planning, and response evaluation, along with prognosis assessment.

**Keywords:** contrast-enhanced ultrasound; portal vein tumor thrombosis; hepatocellular carcinoma





## 1. Introduction

Hepatocellular carcinoma (HCC) is the principal primary liver cancer and one of the most frequent malignant tumors worldwide in patients with chronic liver disease [1,2]. In advanced stages, HCC tends to invade main vascular structures (macroscopic vascular invasion): principally the portal vein system, but also hepatic veins and the inferior vena cava. Advanced stage HCC represents more than 35% of newly diagnosed HCCs [3] and is characterized by vascular invasion or extrahepatic spread; it is included in stages C and D of the Barcelona Clinic Liver Cancer (BCLC) staging system and is often associated with portal vein tumor thrombosis (PVTT), which heavily affects patients' prognosis, with a median survival of 2.7–4 months without therapy, prolonged to 5 months up to 5 years by different treatments and according to liver functional conditions [4–6]. Performance status is poor, ranging from Eastern Cooperative Oncology Group (ECOG) 1–2 in BCLC stage C to 3–4 in BCLC stage D [4]. All the main HCC staging systems (BCLC, TNM, Japan Integrated Staging—JIS, Cancer of the Liver Italian Program—CLIP) recognize the role of vascular invasion as a heavy prognostic factor [7,8].

Portal vein tumor thrombosis (PVTT) is characterized by neoplastic involvement of the portal vein itself or its right or left intrahepatic branches, which can sometimes also externalize to the superior mesenteric vein or splenic veins. Conversely, hepatic vein thrombosis is less frequent than PVTT, but it can lead to consequences that can markedly reduce a patient's survival due to its extension to the inferior vena cava and right atrium [9].

Different groups tried to establish a categorization for PVTT through different classifications in order to define the most appropriate therapeutic path; unfortunately, there is still no clear consensus in the worldwide scientific community on the issue of its treatment.

Ikai et al. identified four grades for PVTT and three for hepatic vein tumor thrombosis: Vp1 with PVTT from distal to second-order branches of the portal vein; Vp2 involving second-order branches directly; Vp3 with PVTT in first-order branches; Vp4 PVTT in the main trunk of the portal vein or invasion of the portal vein branch contralateral to the hepatic lobe where tumor invasion primarily took place; Vv1 with PVTT invading a branch of the hepatic vein; Vv2 with the involvement of the main trunk of hepatic veins; Vv3 PVTT in the right atrium [10].

Cheng et al. proposed another classification consisting of four groups: type $I_0$ regarding microscopic portal invasion; type I resulting from the combination of Vp1 and Vp2; type II including the Vp3 category; type III with PVTT involving the main portal trunk; and type IV involving the superior mesenteric vein [11]. A distinct but closely correlated issue is represented by hepatic vein tumor thrombosis (HVTT). The same group of researchers recently published an article suggesting a new complementary classification regarding hepatic vein neoplastic thrombosis: type I involvement of the hepatic vein, even microvascular infiltration (Ia: engagement of the peripheral hepatic vein; Ib: thrombosis of the main hepatic vein); type II subdiaphragmatic thrombosis or involvement of the retrohepatic section of the inferior vena cava; type III supradiaphragmatic thrombosis (IIIa: in the right atrium; IIIb: the right atrium is not involved) [12]. Li et al. suggested a different classification for hepatic vein neoplastic thrombosis: type I with subdiaphragmatic involvement, type II with supradiaphragmatic involvement but without reaching the atrium, and type III with intra-atrial thrombosis [13]. Another analogue classification, even if less efficient in terms of prognostic prediction, is represented by that suggested by Kokudo et al.: Vv1 with thrombosis in the peripheral hepatic vein, Vv2 involvement of the major hepatic vein, and Vv3 thrombosis in the inferior vena cava [14].

Based on the prognostic implication, it appears necessary to optimize imaging techniques in order to diagnose portal thrombosis as early as possible in patients with HCC.

B-mode ultrasonography (US) is the first-line imaging and is able to detect the presence of portal vein thrombosis but not to distinguish PVTT from non-neoplastic thrombosis. The features of the thrombus consist of a hypoechoic/isoechoic inhomogeneous intravascular mass, creating a partial or complete obstruction of the vessel. The US is not able to identify vascular wall infiltration.

On the other hand, color-doppler US (CDUS) is a low-cost, safe, and non-invasive method that suggests the presence of a malignant thrombus through the identification of intra-thrombotic arterial-like signals with a high resistivity index [15]. It has 66% sensitivity and 100% specificity [16]. Despite the advantage determined by its possibility to provide dynamic information about portal blood flow, unfortunately, it is an operator-dependent method, and it could be impaired by the patient's body habits, intestinal meteorism, or PVTT extension [17], and it is less trustworthy in the staging of mesenteric vein thrombosis [18].

Contrast-enhanced computed tomography (CT) and contrast-enhanced magnetic resonance imaging (MRI) are the principal diagnostic tools for the diagnosis of PVTT, demonstrating the uptake of contrast agents by the thrombotic tissue and thus distinguishing between neoplastic and non-neoplastic PVT [19,20]. Moreover, these radiological techniques are more efficient than CDUS in providing data about thrombosis extension and possible collateral vessels [21]. Different studies demonstrated a better diagnostic performance of gadoxetic acid MRI compared to CT in PVTT detection with a 95% efficacy in the distinction of the nature of the thrombus [22,23]. Both techniques showed similar specificities, but the sensitivity of MRI was higher compared to CT (88–93% versus 70–77%) [24]. The disadvantages associated with CT are the injection of potentially nephrotoxic contrast agents, the application of ionizing radiation, and the absence of dynamic information about portal blood flow [17].

Contrast-enhanced ultrasound (CEUS) is a non-invasive, cheap, and easily repeatable method with a good safety profile that can also be used in patients with impaired renal function. It allows real-time observation of the behavior of an organ or lesion with the administration of a contrast agent. It represents an important tool for the identification of PVTT, particularly differentiating neoplastic and non-neoplastic thrombosis through the analysis of ultrasound enhancement characteristics of the thrombosis (malignant findings are characterized by intraluminal arterial hyperenhancement during the arterial phase and washout in the portal or late phase, while benign thrombosis lacks contrast enhancement in any phase), thus allowing a more refined disease staging, appropriate treatment planning, and response evaluation, together with prognosis assessment [25]. It is also able to detect small arterial vessels in the thrombotic tissue, determining real-time arterial enhancement with the typical arterial intermittent pulsation [26]. Due to its high sensitivity, specificity, and accuracy, it is recommended for PVTT detection by the guidelines of the European Association for the Study of the Liver (EASL), the European Federation of Societies for Ultrasound in Medicine and Biology (EFSUMB), and the World Federation for Ultrasound In Medicine and Biology (WFUMB) [24,27–29].

It is important to underline that CEUS is more efficient in distinguishing small portal thrombi that can be missed by CT scans due to its imperfect ability to differentiate the principal neoplastic mass from its intravascular branch [30].

Even 18F-Fluorodeoxyglucose positron emission tomography/CT (18F-FDG PET/CT) was applied in the differential diagnosis between benign and malignant portal thrombosis, with the possibility of detecting metabolic anomalies in PVTT due to its maximum standardized uptake value (SUVmax) being superior to 3.35 as reported by Hu et al. [31].

The ultimate diagnostic tool is represented by percutaneous US-guided fine-needle biopsy, which is the most effective method to distinguish between neoplastic and non-neoplastic thrombotic tissue, particularly in those cases in which the previous tools could not achieve a satisfying result but had the disadvantage of being an invasive method [32].

The aim of our review is to critically revise the state of the art regarding CEUS performance in the diagnosis of PVTT.

## 2. Radiological Footsteps towards the Diagnosis of Neoplastic Portal Vein Thrombosis

The first imaging modality for the detection of PVTT is notably CDUS, thanks to its high accuracy in detecting thrombosis that involves the PV trunk and its intrahepatic branches. Pozniak et al. first detected pulsatile flow within a portal thrombus by CDUS [33]. Subsequently, several studies underlined the importance of CDUS as a diagnostic instrument in the distinction between non-neoplastic and neoplastic PVT, with good results in terms of sensitivity, specificity, and accuracy.

The study by Tanaka et al. compared CDUS and hepatic angiogram findings on 40 patients with HCC who were divided into two groups (18 with PVT and 22 without PVT) and aimed to determine the efficacy of CDUS in PVTT detection. They documented the presence of color signals within the thrombus (due to tumoral neovascularity), underlining the role of CDUS in both portal vein screening and the identification of intravascular neoplastic spread. The presence of pulsative flow in a thrombus localized in the main portal vein was pathognomonic for PVTT (89% sensitivity, 100% specificity, and 96% accuracy) [34]. Despite these appealing results, the reduced dimension of the study population hinders the application of the proposed methods in the diagnostic pathway of PVTT.

Similarly, Lencioni et al. analysed CDUS in 13 patients with PVTT and reported the presence of pulsatile intra-thrombotic arterial flow in 12/13 patients with neoplastic thrombosis and the absence of this sign in subjects with benign thrombosis, stating the reliability of CDUS in the differentiation of malignant and benign thrombosis (sensitivity 92% and specificity 100%). Unfortunately, the reduced dimension of this study sample impairs the possibility of fully validating its role in PVTT detection [35].

In order to improve the sensitivity of US in the detection of PVTT, the role of CEUS has been evaluated in different cohorts. A study performed on 56 cirrhotic patients demon-

strated the usefulness of CEUS compared to conventional CDUS in the differential diagnosis of benign and malignant PVT: the ability of CEUS to detect the arterial flow within the thrombus was higher than that of CDUS (94% sensitivity and 100% specificity for CEUS vs. 57% sensitivity and 95% specificity for CDUS) [36].

The retrospective study by Marshall et al. highlighted the role of CEUS (performed through intravenous injection of the first generation contrast agent Levovist) as a cheap and simple method complementary to standard screening for PVT to perform before liver transplantation (LT): they examined 33 LT candidates with an inadequate assessment of portal venous flow by CDUS and found that CEUS improved portal vein visualization in 94% of patients, with 87% overall diagnostic accuracy and an increase in median diagnostic confidence from 50% to 90% after contrast agent administration ($p < 0.001$) [37].

Similar results were obtained in the prospective study by Rossi et al., who analyzed at baseline and after three months 233 patients with portal or hepatic vein thrombosis from 316 patients with malignant liver tumors (220 HCCs, 14 cholangiocarcinomas, and 82 hepatic metastases from different solid tumors); at baseline, they identified 79 patients with thrombosis and after three months, 83 [26]. Malignant thrombosis was found in 81 patients (97.6%) among those with portal thrombosis (83 subjects). Using US-guided biopsies as the gold standard for PVTT diagnosis, they demonstrated the superiority of CEUS (using Sonovue as a contrast agent) over US ($p = 0.058$, with borderline results) and CDUS ($p = 0.004$) for the detection of portal and hepatic vein thrombosis; for the characterization of the thrombosis, CEUS was more sensitive than US ($p = 0.02$) and CDUS ($p < 0.0005$).

The high accuracy of CEUS in the detection of PVTT has also been underlined by Tarantino et al., who demonstrated the superior sensitivity of CEUS (performed with Sonovue, Bracco, Italy) not only over CDUS but also over fine needle biopsy (FNB) in a group of 54 cirrhotic patients with biopsy-proved HCC (sensitivity: 88%, 20%, and 76% for CEUS, CDUS, and FNB, respectively) [10]. The highest accuracy was achieved by CEUS compared to CDUS and FNB (92.5%, 50%, and 78.7%, respectively). During the follow-up, 34 subjects developed malignant thrombosis: CEUS was able to identify 30 cases of PVTT, whereas CDUS and FNB only identified 7 and 19 cases, respectively. No false-positive results were detected with the three methods.

Notably, the last two studies challenged the results of previous reports concerning CDUS sensitivity in the detection of PVTT, showing that it is highly dependent on the size of the thrombus [10,26].

These results were confirmed in the retrospective study by Chammas et al. performed on a group of 43 cirrhotic patients with HCC and PVT: the accuracy of CEUS and CDUS in differentiating benign from malignant thrombosis was compared using serial imaging follow-up by CT and/or MRI for at least 6 months after initial examination [38]. The only CDUS criterion for defining PVTT was the presence of an arterial flow pattern inside the thrombus; on the contrary, criteria for benign PVT were the absence of blood flow inside the thrombus or the presence of a venous blood flow pattern. CEUS criteria to define PVTT were the presence of arterial phase enhancement inside the thrombus. Overall, CDUS detected venous blood flow (or its total absence) in 13 of the 21 (62%) benign PVTs and arterial flow in 2 of the 22 (9.1%) malignant PVTs. On the contrary, CEUS found blood flow in 21 of 21 (100%) cases of benign PVT and in 20 of 22 (90.9%) cases of malignant PVT, all of which exhibit earlier arterial blood flow than benign thrombi (sensitivity, specificity, and accuracy of CEUS: 90.9%, 100%, and 95.3%, respectively vs. 9.1%, 100%, and 53.4% of CDUS, respectively) [38].

The performance of CEUS compared to standard radiological imaging has also been tested in a retrospective study on 50 patients affected by HCC [39]. In this case, PVT was previously identified on CT/MRI imaging, and CEUS was performed within 4 weeks of diagnosis. Once more, CEUS demonstrated good accuracy in defining the kind of thrombus, with 100% sensitivity, 87% specificity, 95% positive predictive value, and 100% negative predictive value. Furthermore, arterial phase enhancement emerges as the most useful

CEUS qualitative parameter for discriminating between non-neoplastic and neoplastic thrombosis (Figure 1). Similar results were obtained by Song et al. [40] and Rossi et al. [41]. In particular, the last study demonstrated a better performance of CEUS (determined by the low mechanical index) compared to CT scan in PVT detection ($p < 0.0001$) and thrombus characterization ($p = 0.0001$) both for sensitivity (98% CEUS, 67.6% CT) and specificity (100% CEUS, 60% CT) in 50 HCC patients with biopsy-proven PVTT.

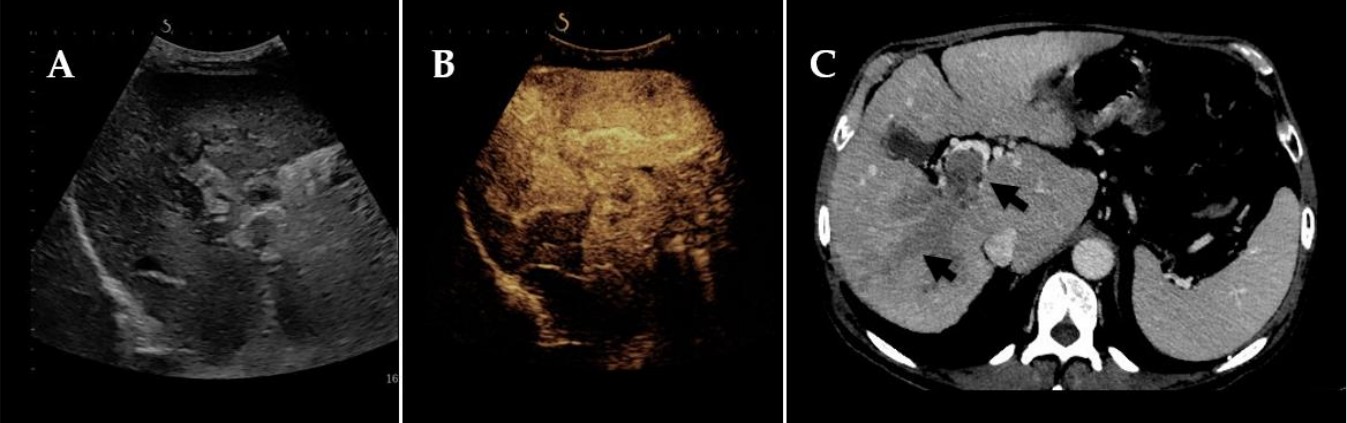

**Figure 1.** Neoplastic portal vein thrombosis in a 73-year-old patient with advanced hepatocellular carcinoma, detected through different imaging techniques: (**A**) B-mode ultrasound; (**B**) Contrast-Enhanced Ultrasound (arterial phase); (**C**) Computed Tomography.

A recent meta-analysis by Chen et al. confirmed CEUS as an excellent tool for PVT differentiation in 425 HCC patients, electing CEUS as a concrete alternative or a substitute for CT and/or MRI with a pooled sensitivity of 0.94 (95%CI, 0.89–0.97), a specificity of 0.99 (95%CI, 0.80–1.00), and an area under the receiver operating characteristic curve (AUROC) of 0.97 (95%CI, 0.95–0.98) [27].

The accuracy of CEUS in the characterization of PVT has also been compared to FNB by Sorrentino et al. in a group of 186 patients with HCC and PVT [42]. CEUS, demonstrating a progressive hypoenhancement in non-neoplastic thrombosis and an early homogeneous enhancement in PVTT, had higher accuracy than biopsy. The latter, in fact, may give false-negative results due to casual sampling of the thrombus. The authors concluded that CEUS itself could guide FNB thrombus sampling when cytology confirmation of malignant thrombosis is needed.

Finally, recent studies evaluated the role of dynamic CEUS based on objective quantitative parameters in PVTT diagnosis. In the study by Li et al., the PVTT perfusion curve was defined as "fast-up and fast-down", and statistically significant differences were observed between PVTT and surrounding liver parenchyma regarding time to peak (TTP), arrival time (AT), rise time (RT), peak intensity (PI), rising slope rate (RSR), and washout time (WT) [43]. As demonstrated for qualitative CEUS, the sensitivity, specificity, and accuracy of dynamic CEUS for the diagnosis of PVTT were comparable with those of CT (97.8% versus 96.7%, 90.2% versus 86.4%, and 100% versus 97.7%, respectively), with an AUROC of 0.939 and 0.933, respectively.

Both qualitative and dynamic CEUS were tested by Kwon et al. in a prospective study on 49 patients with HCC and compared to Diffusion-Weighted MRI (DW-MRI) in differentiating benign and malignant PVT [44]. The authors demonstrated that all qualitative CEUS parameters (enhancement on the arterial phase, washout on the portal phase, vessel occlusion, and vessel expansion) were significantly more common in malignant thrombosis than in benign thrombosis ($p < 0.05$). As concerning dynamic CEUS, larger Area Under the Curve (AUC), prolonged Full Width at Half Maximum (FWHM, time between half amplitude values on each side of the maximum), and shorter RT (Rise Time, that is, time from injection until the peak of enhancement) were observed in malignant thrombosis. However, due to the insufficient accuracy of FWHM and the poor intra-observer agreement

of RT, only AUC was proposed as a valid parameter for the diagnosis of PVTT. Overall, CEUS diagnostic accuracy in discriminating thrombus nature was similar to that of DWI-MRI as for sensitivity, specificity, and accuracy (all above 90%) with a strong correlation between the two techniques ($p < 0.0001$), thus supporting the role of CEUS as a valid and promising tool in PVT differentiation (91.4% sensitivity, 100% specificity, 93.9% accuracy, 100% positive predictive value, and 82.4% negative predictive value) [43].

A potentially promising role of CEUS was suggested in the prospective study by Sparchez et al., who proposed CEUS as a safe technique for guiding core biopsy in cases of PVTT and occult HCC. In those cases where the primary liver lesion is not evident, the authors observed that a "coarse echo pattern" can be identified in the parenchyma close to the neoplastic thrombus and is usually the manifestation of a missing tumor. For this study, the global sensitivity of biopsy in the detection of PVTT was 94.4%, with a sensitivity of 100% in the CEUS-guided biopsy and 92.8% in case of the US-guided procedure. The diagnosis of primary HCC was achieved in 6 of 10 subjects that underwent CEUS-guided parenchymal biopsy [45].

Table 1 summarizes the performance achieved by CEUS in the detection and characterization of PVTT compared to other imaging techniques and FNB.

A retrospective study by Burciu et al. on patients with liver cirrhosis, HCC, and PVT assessed the efficacy of the combination of CEUS and biological tests in the identification of PVT (contemporarily confirmed by CT and MRI imaging). They noticed that CEUS was reliable for the differentiation of malignant and benign PVT, with a sensitivity of 98.6% and a specificity of 89.3%. When CEUS is associated with alpha-fetoprotein (AFP), its diagnostic power for the differentiation of benign and malignant PVT is increased (AFP < 20 ng/dL and AFP > 200 ng/dL, respectively). Among those with PVTT, 100% were correctly classified as having AFP > 200 ng/dL, and this value was established as the diagnostic value to rule-in PVTT. In this way, they created a score for PVT classification, with a better performance than CEUS or AFP alone (respectively: AUC −0.99 vs. −0.93, $p = 0.025$; AUC −0.99 vs. −0.96, $p = 0.047$). Despite the promising results, the author identified some notable weaknesses: the number of overall patients was quite small, the group of those with HCC was limited, and the study was monocentric. Furthermore, CEUS depends on the operator's ability (in the study, they underline the lack of estimation of interoperator reproducibility); a second evaluation is advisable to increase CEUS reliability; diagnostic efficacy was not evaluated objectively through computer-assisted devices. Another issue was the lack of AFP level records for each of the patients included in the study, thus further reducing the sample of those that could be considered in this analysis [46].

An innovative perspective is offered by Wang et al. with the quantitative analysis of blood perfusion through quantification software that could potentially play a relevant role in the early prediction of PVTT. They confronted CEUS performed with Sonovue on 24 subjects with HCCs and PVTT and on 48 patients with HCC without PVTT (control group). In physiological conditions, blood perfusion of the liver is granted mainly by portal flow (75%), and slightly by arterial blood (25%). PVTT produces noteworthy alterations in liver perfusion that are normally determined by portal blood flow and arterial flow. The subsequent dynamic analysis was performed afterwards with specific software that extrapolates quantitative hemodynamic data. They observed that in the case of PVTT, the subsequent alteration of liver blood inflow determined significantly faster rising time (RT) and TTP faced than in the control group. Conversely, the intensity maximum (IMAX), which normally should be proportional to contrast agent concentration, was lower in the PVTT group because of the arterial supply compensation following PVTT [47].

Even if all these studies provide relevant data that could enhance CEUS's role as the concrete milestone for PVTT diagnosis, most of them regard small groups of patients and the retrospective design of some of these patients, thus reducing their statistical power.

**Table 1.** Performance of CEUS compared to other imaging techniques and FNB in the detection and characterization of PVTT.

| | Patients | Imaging Techniques | Sensitivity | Specificity | Accuracy | PPV | NPV |
|---|---|---|---|---|---|---|---|
| Ricci et al. [36] | 56 (46 benign PVT 16 malignant PVT) | CDUS CEUS | 57% 94% | 95% 100% | NA | NA | NA |
| Lencioni et al. [35] | 19 (6 benign PVT; 13 malignant PVT) | CDUS | 92% | 100% | 95% | NA | NA |
| Tanaka et al. [34] | 18 PVT 22 no PVT | CDUS | 89% | 100% | 96% | NA | NA |
| Tarantino et al. [10] | 54 | CEUS CDUS FNB | 88% 20% 76% | 100% 100% 100% | 92.5% 50% 78.7% | 100% 100% 100% | 83.3% 42.5% 33.3% |
| Chammas et al. [38] | 43 (21 benign PVT; 22 malignant PVT) | CEUS CDUS | 90.9% 9.1% | 100% 100% | 95.3% 53.4% | NA | NA |
| Song et al. [40] | 17 | CEUS | 100% | 66.7% | 93.3% | NA | NA |
| Raza et al. [39] | 50 | CEUS | 100% | 83–92% | NA | 95–97% | 100% |
| Rossi et al. [41] | 50 | US CDUS CEUS CT | 86.4% 54.3% 98% 67.6% | - - 100% 60% | NA | NA | NA |
| Li et al. [43] | 93 | CEUS CT | 97.8% 96.7% | 90.2% 86.4% | 100% 97.7% | NA | NA |
| Kwon et al. [44] | 49 | CEUS DWI-MRI | 91.4% 90.6% | 100% 100% | 93.9% 93.2% | 100% 100% | 82.4% 80% |
| Sorrentino et al. [42] | 186 | CEUS FNB | 89.6% 89.6% | 100% 100% | ND | 100% 100% | 89.2% 89.2% |

Abbreviations: CDUS—colour doppler ultrasound; CEUS—contrast-enhanced ultrasound; CT—computed tomography; DWI-MRI—Diffusion-Weighted Magnetic Resonance Imaging; FNB—fine needle biopsy; NA—not available; NPV—negative predictive value; PPV—positive predictive value; PVT—portal vein thrombosis; US—ultrasound.

## 3. Discussion and Future Perspectives

An accurate detection of PVTT is crucial for both the correct assessment of HCC extension and the choice of the most adequate treatment. Biopsy is still the main tool for a definitive diagnosis, despite its invasiveness, but it should be avoided in particular conditions (e.g., presence of ascites, impaired coagulation due to chronic liver disease) and, moreover, it could be affected by sampling error.

CEUS is superior to traditional radiological diagnostic imaging (CT and MRI) in PVTT detection for several reasons: its high sensitivity allows it to identify the signals sent by the contrast microbubbles, which have a size similar to red blood cells. Their resonant oscillatory behavior is stable, and they remain exclusively within the blood vessels, reflecting the blood volume in a specific field during each enhancement phase [48]. Conversely, a CT-scan iodine-based contrast agent can spread through endothelial cells into interstitial tissue, originating the "interstitial phase" of enhancement; in fact, due to neoplastic neovascularization, the endothelium acquires an increased permeability, determining a faster diffusion in the interstitial district, with a consequent dimming of venous washout during the portal phase [49,50].

Among its advantages, CEUS allows the detection of smaller neoplastic thrombi due to its ability to identify intrathrombotic neovascularization, eliminating the need for a biopsy

in most controversial cases and overcoming the possibility of false negatives determined by CDUS examination alone.

CEUS provides a fast real-time examination; it takes about 3–4 min in total, reaching a competitive result compared to other traditional radiological techniques.

Moreover, it is an extremely valuable tool for patients with contraindications to CT or MRI contrast agents (e.g., chronic kidney impairment, allergies).

However, CEUS is a poorly standardized method, depending on the operator's experience and skill, the quality of the available equipment to perform the examination, and the physical characteristics of the patient (body habitus, abdominal scars from previous surgical interventions), respiratory movement artifacts (due to underlying pathologies), and the scarce acoustic window due to meteorism.

Furthermore, it is important to notice that vascular abnormalities that are common in cirrhotic livers (e.g., arterio-venous and arterio-portal shunts, aneurysms), along with the hypertrophy of the arterial feeding of the thrombus, could interfere with the detection of the maximum perfusion of the thrombus, thus reducing CEUS diagnostic power.

Despite some disadvantages presented by CEUS, its real-time performance grants the acquisition of qualitative and quantitative information during the arterial, portal, and late phases, including the precise moment of peak enhancement in the arterial phase.

An added value could be given by dynamic-CEUS analysis: thanks to the possibility to examine quantitatively every stage from the beginning of contrast agent enhancement to its peak and subsequent washout pattern, dynamic-CEUS provides a more objective evaluation of the target behavior during the exam.

It is advisable that, in the future, CEUS and CT/MRI may become complementary tools, not only for the detection of PVTT but also as a follow-up method during HCC treatment. In fact, CEUS represents a cheap, easily repeatable method that could be applied precociously during follow-up schedules. The absence of ionizing radiation gives the possibility of repeating it with an increased frequency for CT/MRI.

## 4. Conclusions

Traditional imaging is not always accurate in PVTT detection, while CEUS proves to be a precise and accessible tool. Due to its sensitivity and specificity, it can be considered an excellent substitute for CT and MRI, particularly after inconclusive CR/MRI, with the unvaluable advantages of avoiding ionizing radiation and nephrotoxicity. It could acquire increasing importance in the evaluation of undetected PVTT in candidates for liver transplantation. Further studies are necessary to explore all the potentially valuable applications of CEUS.

**Author Contributions:** L.C., M.E.A. and M.A.Z. designed the review; L.C., S.D.F. and L.G. performed the scientific literature research; L.C., M.E.A., S.D.F. and L.G. drafted the manuscript. L.C., M.E.A. and M.A.Z. revised the manuscript; M.A.Z. and A.G. revised the manuscript critically for intellectual content. All authors have read and agreed to the published version of the manuscript.

**Funding:** This research received no external funding.

**Institutional Review Board Statement:** Not applicable.

**Informed Consent Statement:** Not applicable.

**Data Availability Statement:** Not applicable.

**Acknowledgments:** Thanks to Fondazione Roma for the continuous support of our scientific research.

**Conflicts of Interest:** The authors declare no conflict of interest.

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
