# Peer review of "The Role of Contrast-Enhanced Ultrasound (CEUS) in the Detection of Neoplastic Portal Vein Thrombosis in Patients with Hepatocellular Carcinoma"

_tomography, doi:10.3390/tomography9050154_

Round 1

Reviewer 1 Report

This review paper discusses using contrast-enhanced ultrasound (CEUS) to detect portal vein tumor thrombosis (PVTT) in hepatocellular carcinoma patients. It compares CEUS to other imaging modalities and shows CEUS has higher accuracy for diagnosing PVTT through its ability to characterize blood flow patterns. The paper concludes CEUS is a valuable tool for PVTT detection that can improve HCC staging and treatment planning.

= Strengths:

-Clearly explains the importance of accurately detecting PVTT in hepatocellular carcinoma patients for prognosis and treatment planning.

-Provides a good overview of different imaging techniques for detecting PVTT, including ultrasound, CT, MRI, and CEUS.

-Comprehensively reviews many studies comparing CEUS to other imaging modalities in PVTT detection, demonstrating it has high sensitivity and specificity.

-Explains the qualitative and quantitative enhancement patterns seen on CEUS that can distinguish malignant vs benign thrombosis.

-Discusses some promising new techniques like dynamic CEUS analysis and perfusion quantification software.

-Makes a good case for CEUS as a first line imaging modality for PVTT detection, complementing CT/MRI.

=However, several critiques need to be concisely addressed:

-The introduction could provide more, concise background on the pathology and classification systems for PVTT.

-When reviewing studies, more, concise details on patient cohorts, CEUS techniques, and statistics would strengthen the evidence.

-More discussion on limitations of CEUS like operator dependence, artifacts, and inter-observer variability would add balance.

-Doesn't really critique the quality of the studies reviewed or identify limitations/gaps in the current literature.

-Conclusions are a bit repetitive from the rest of the paper, could be condensed and focused on future directions.

none.

Author Response

This review paper discusses using contrast-enhanced ultrasound (CEUS) to detect portal vein tumor thrombosis (PVTT) in hepatocellular carcinoma patients. It compares CEUS to other imaging modalities and shows CEUS has higher accuracy for diagnosing PVTT through its ability to characterize blood flow patterns. The paper concludes CEUS is a valuable tool for PVTT detection that can improve HCC staging and treatment planning.

Strengths:

-Clearly explains the importance of accurately detecting PVTT in hepatocellular carcinoma patients for prognosis and treatment planning.

-Provides a good overview of different imaging techniques for detecting PVTT, including ultrasound, CT, MRI, and CEUS.

-Comprehensively reviews many studies comparing CEUS to other imaging modalities in PVTT detection, demonstrating it has high sensitivity and specificity.

-Explains the qualitative and quantitative enhancement patterns seen on CEUS that can distinguish malignant vs benign thrombosis.

-Discusses some promising new techniques like dynamic CEUS analysis and perfusion quantification software.

-Makes a good case for CEUS as a first line imaging modality for PVTT detection, complementing CT/MRI.

However, several critiques need to be concisely addressed:

-The introduction could provide more, concise background on the pathology and classification systems for PVTT.

Re: According to reviewer suggestion we added in the text more informations concerning the pathology and classification systems for PVTT (see lines 41-72).

-When reviewing studies, more, concise details on patient cohorts, CEUS techniques, and statistics would strengthen the evidence

Re: According to reviewer suggestion, we have revised the text (see lines 46-72, 157, 164-165, 167-173, 178-179, 237-238, 253)

-More discussion on limitations of CEUS like operator dependence, artifacts, and inter-observer variability would add balance

Re: According to reviewer suggestion, we have additionally pointed out this aspect in the text  (see lines 329-337)

-Doesn't really critique the quality of the studies reviewed or identify limitations/gaps in the current literature

Re: Following the suggestion of the reviewer we have better clarified this aspect in the text (see lines 302-304).

-Conclusions are a bit repetitive from the rest of the paper, could be condensed and focused on future directions à

Re: Following the suggestion of the reviewer we have rearranged the section dividing it in two parts “Discussions and Future perspectives” and “Conclusions”.

Reviewer 2 Report

Introduction

Detailed and significant information for the purpose of the review.

-In Line 57, in Cheng classification, type I0 is the correct one instead of I0.

2. Radiological footsteps

A comprehensive review of available studies.

-Line 163, Rossi study must be cited as [26] instead of [38], as in References.

-Line 213, Sorrentino study must be cited as [42] instead of [27], as in References.

-In Table 1, 21 benign PVT, 22 malignant PVT were actually in Chammas et al study [38] (row in table 1 immediately below) and not in Tarantino study.

3. Conclusions

- Although information included into the section are pertinent and important, the chapter included in Conclusions  are usually more concise; some information may be moved to the previous section or in a new section entitled Discussions.

Additional comments: a very comprehensive review of current data regarding the role of CEUS in the diagnosis of PVT, including the significance of qualitative and dynamic criteria for CEUS in the PVT and complementary role to CT and IRM for the diagnosis.  

In Lines 214-215, the term "precocious homogenous enhancement" may be replaced with "early homogenous enhancement"

Author Response

Introduction

Detailed and significant information for the purpose of the review.

-In Line 57, in Cheng classification, type I0 is the correct one instead of I0.

Re: We changed accordingly (it was a typing error).

  1. Radiological footsteps

A comprehensive review of available studies.

-Line 163, Rossi study must be cited as [26] instead of [38], as in References.

Re: We changed accordingly.

-Line 213, Sorrentino study must be cited as [42] instead of [27], as in References.

Re: We changed accordingly.

-In Table 1, 21 benign PVT, 22 malignant PVT were actually in Chammas et al study [38] (row in table 1 immediately below) and not in Tarantino study.

Re: We changed accordingly (it was a typing error).

  1. Conclusions

- Although information included into the section are pertinent and important, the chapter included in Conclusions  are usually more concise; some information may be moved to the previous section or in a new section entitled Discussions.

Re: Following the suggestion of the reviewer we have rearranged the section dividing it in two parts “Discussions and Future perspectives” and “Conclusions”.

Additional comments: a very comprehensive review of current data regarding the role of CEUS in the diagnosis of PVT, including the significance of qualitative and dynamic criteria for CEUS in the PVT and complementary role to CT and IRM for the diagnosis.  

Comments on the Quality of English Language

In Lines 214-215, the term "precocious homogenous enhancement" may be replaced with "early homogenous enhancement".

Re: We changed accordingly.